# Position: Current Model Licensing Practices are Dragging Us into a Quagmire of Legal Noncompliance

**Moming Duan** [* 1]  **Mingzhe Du** [* 1]  **Rui Zhao** [2]  **Mengying Wang** [3]  **Yinghui Wu** [3]  **Nigel Shadbolt** [2]  **Bingsheng He** [4]

## Abstract

The Machine Learning (ML) community has witnessed explosive growth, with millions of ML models being published on the Web. Reusing ML model components has been prevalent nowadays. Developers are often required to choose a license to publish and govern the use of their models. Popular options include Apache-2.0, OpenRAIL (Responsible AI Licenses), Creative Commons Licenses (CCs), Llama2, and GPL-3.0. Currently, no standard or widely accepted best practices exist for model licensing. But does this lack of standardization lead to undesired consequences? Our answer is Yes. After reviewing the clauses of the most widely adopted licenses, we take the position that *current model licensing practices are dragging us into a quagmire of legal noncompliance*. To support this view, we explore the current practices in model licensing and highlight the differences between various model licenses. We then identify potential legal risks associated with these licenses and demonstrate these risks using examples from real-world repositories on Hugging Face. To foster a more standardized future for model licensing, we also propose a new draft of model licenses, ModelGo Licenses (MGLs), to address these challenges and promote better compliance. https://www.modelgo.li/

## 1. Introduction

With the emergence of Parameter Efficient Fine-Tuning (PEFT) (Hu et al., 2022; Pfeiffer et al., 2020) technologies, developers can now customize Pre-Trained Models (PTMs) (Jiang et al., 2023) to address various downstream tasks at an affordable computational and data resources. By the end of 2024, over 1.2 million models have been publicly published on Hugging Face (HF), distributed and used under various licenses (or aggrements). These licenses can be grouped into three categories: 1) Open Source Software (OSS) licenses (Rosen, 2005), such as Apache, MIT, and GPL; 2) Free-content (or dataset) licenses, such as CCs (Commons, 2024) and PDDL; 3) Model-specific licenses, such as OpenRAILs (Contractor et al., 2022), Llama2 Community License (Meta Platforms, 2024), and Gemma Terms of Use (Google, 2024). In current licensing practices, publishers are free to declare any of these licenses based purely on personal preference. For example, C4AI Command R7B adopts CC-BY-NC-4.0 license, which prohibits commercial use of licensed materials and their derivatives. CKIP-Transformers uses GPL-3.0 license to align with its accompanying code repository. However, upon closer review of the terms and conditions in these licenses, we have identified the following important issues about model licensing.

The first issue is **License Mismatch**, OSS and free-content licenses were not designed for model publishing. This means their definitions and clauses may not be well-suited for the context of ML. For example, CC licenses explicitly state that *"Licensed Material is a musical work, performance, or sound recording"*. Thus, when applying CC licenses to model weights (or code[1]), it becomes ambiguous to interpret concepts like *translated, altered, arranged, transformed*. On the other hand, these mismatched licenses also lack governance over ML technologies such as knowledge transfer (You et al., 2021) and Mix-of-Experts (MoE) (Jacobs et al., 1991), and fail to provide dispute resolution mechanisms regarding the ownership of Intellectual Property (IP) rights for derivatives and generated content.

The second issue is **License Proliferation**. Legal noncompliance can occur if developers create derivatives (e.g., via quantization or fine-tuning) based on a model under copy-

---

[*]Equal contribution [1]Institute of Data Science, National University of Singapore [2]Department of Computer Science, University of Oxford, Oxford, UK [3]Department of Computer and Data Sciences, Case Western Reserve University, Cleveland, USA [4]School of Computing, National University of Singapore. Correspondence to: Moming Duan <moming@nus.edu.sg>.

*Proceedings of the $42^{nd}$ International Conference on Machine Learning*, Vancouver, Canada. PMLR 267, 2025. Copyright 2025 by the author(s).

[1]Can I apply a Creative Commons license to software?

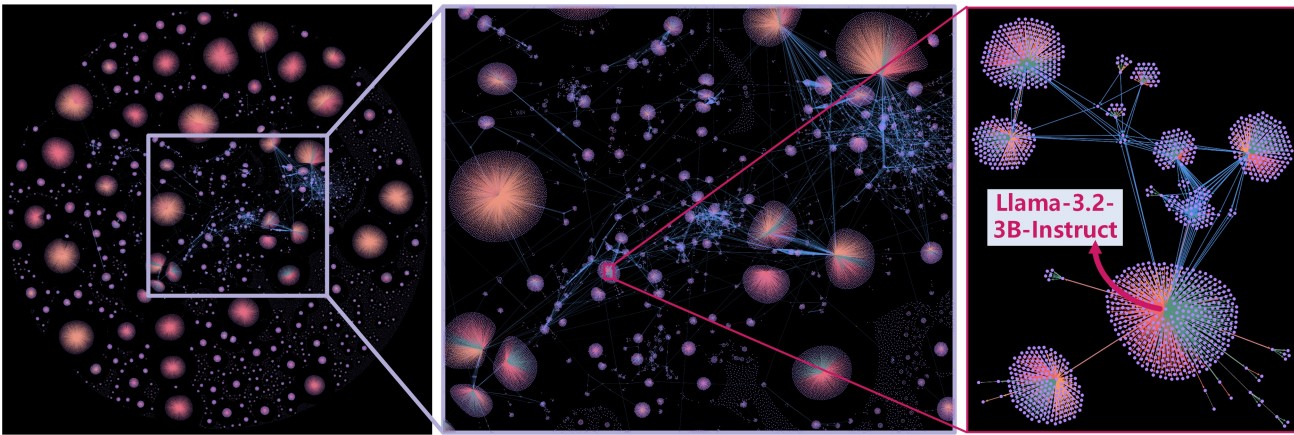

*Figure 1.* Visualization of Model Dependencies on HF. Purple nodes represent models, and edges represent model dependencies: **Finetune**, **Adapter**, **Quantization**, and **Merge**. **Left**: Global view, showing the widespread practices of model reuse. **Middle**: Zoomed-in view, where blue lines between models indicate they are merged into new models. **Right**: Subgraph view, illustrating the 3-hop neighbor models and their dependencies related to meta-llama/Llama-3.2-3B-Instruct.

left licenses like GPL[2] and CC-BY-SA[3] but republish the derivatives under a different license. License proliferation is a well-known issue in the OSS field (Gomulkiewicz, 2009), and various methods have been developed to identify license information, such as scanning for SPDX identifiers[4] in source files or matching file provenance with external databases (German et al., 2010; Ombredanne, 2020). However, widely used model file formats, such as Safetensors, GGUF, and OpenVINO IR, primarily store model weights (and, in some cases, model architecture)[5], making license scanning no longer effective. Furthermore, existing OSS license analysis tools do not yet support emerging model licenses. More fundamentally, an unresolved issue persists: what types of ML reuse methods might trigger license proliferation? These status quo and questions constitute the unique challenges of legal compliance in ML.

The last issue is **License Conflict**. Simply put, components with incompatible licenses cannot coexist in a project (if their exclusive terms are both triggered). For example, GPL-licensed material cannot contain components under licenses with discriminatory terms (Greenbaum, 2016), such as non-commercial restrictions in CC-BY-NC. The common solution in OSS projects is to identify such conflicts based on known incompatible license pairs[6] (Cui et al., 2023).

While scanning licenses for model components presents its own challenges, we can still encode known license conflicts into a checklist. For example, RAILs and Llama2 License, which also contain discriminatory terms, could conflict with GPL-3.0 (Contractor et al., 2022).

Even though it is possible to exploit solutions from OSS projects to address the issues above, one unique nature of ML projects, **Implicit Dependency**, makes them significantly harder to resolve. Unlike OSS projects, where dependencies typically involve file inclusion, code snippet copying, or library linking[7], ML models can have implicit dependencies by learning representations from other models without substantially copying any part of those models (e.g., Alpaca (Taori et al., 2023) fine-tuned from LLaMA (Touvron et al., 2023) with instructions generated by GPT-3.5). This means it is difficult to identify the complete supply chain of a model (visualized in Figure 1). Worst still, some model-specific licenses have terms that can be triggered by such implicit dependencies. For instance, Llama2 License clause 1.v reads: "*You will not use the Llama Materials or any output or results of the Llama Materials to improve any other large language model ...*". Such a copyleft-style clause can be propagated to other non-Llama2 projects through implicit dependencies, potentially causing license conflicts with GPL-licensed projects.

In this paper, we point out the dilemmas in current model licensing practices: 1) A license is chosen but may not be suitable to clarify model publishing policies; 2) The chosen license may also be noncompliant due to license conflicts; 3) Resolving such conflicts is challenging because

---

[2]GPL-3.0 Clause 5(c): *You must license the entire work, as a whole, under this License to anyone ...*

[3]CC-BY-SA-4.0 Clause b1: *The Adapter's License you apply must be a Creative Commons license with the same License Elements, this version or later, or a BY-SA Compatible License.*

[4]SPDX License List

[5]ONNX offers the possibility to store license information within the model file.

[6]GPL-Incompatible Free Software Licenses

[7]Black Duck® software composition analysis

it is difficult to identify all dependencies. Ultimately, this leads us to suspicion of license violations. Therefore, we take the position that **current model licensing practices are dragging us into a quagmire of legal noncompliance**. We will use in-the-wild repositories to demonstrate our position and propose feasible solutions to alleviate the mentioned risks. The contributions are:

- We explore the legal compliance challenges in model licensing through an analysis of ML repositories in the wild, spanning 15K+ models and 14K+ dependencies.

- We qualitatively review widely used licenses and highlight their misinterpretation risks in the ML context.

- We propose a set of model licenses, MGLs, to advocate for standard model licensing in the ML community.

## 2. ML Model Reuse and Licensing

We briefly introduce the background and related studies in this section. We illustrate the status quo of model licensing practices using statistics derived from real-world cases. Specifically, we collected the model cards (Mitchell et al., 2019) of the top 8K downloaded models on HF and crawled their model trees, resulting in a graph with 151,302 models and 146,818 relationships (accessed December 2024). A visualization via the Neo4j Browser[8] of our collected data is shown in Figure 1. Each model is represented as a node, and their relationships (model dependencies) are depicted as edges with different colors: Finetune (50%), Adapter (33%), Quantization (10%), and Merge (4%)[9]. All statistical results presented in this paper are derived from this dataset unless stated otherwise.

### 2.1. How ML Models Are Licensed?

A license is a legal agreement between the licensor and the licensee that governs the use and distribution of licensed materials. It typically includes definitions of terms, clauses specifying granted and reserved rights, restrictions on particular behaviors, liability, conditions for termination, and other provisions. Before publishing a model to the web, it is advisable (though not mandatory) to adopt a license to limit the scope of subsequent use, for example, prohibiting commercial use or hacking behaviors (Contractor et al., 2022). Figure 2 presents the pie chart of top-10 license choosing preferences. As shown in the figure, Apache and MIT are the most widely adopted licenses, both of which are OSS licenses. Additionally, 30% of models (labeled as *null*) lack a declared license, leaving them vulnerable to misuse and potential IP theft. Furthermore, 6.2% of models (labeled as

[8]Neo4j Graph Database & Analytics

[9]For clarifications of these labels, see Section 3.2. The reversed relationships, Base Model (2.2%), are hidden.

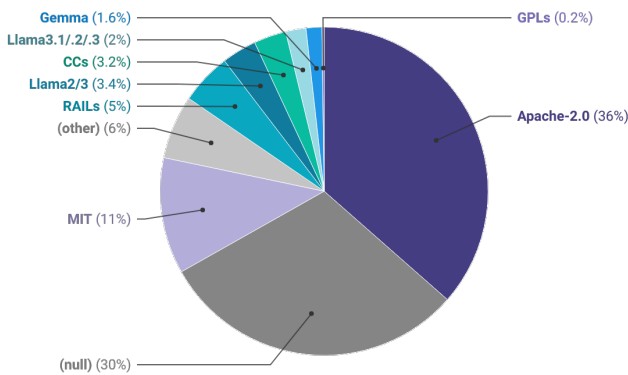

*Figure 2.* Top-10 License Preferences for Model Publishing. Some licenses are grouped into GPLs, CCs, RAILs; *null*: no license claimed; *other*: unclassified (or custom) license claimed.

*other*) use an unclassified license, typically a custom one. Other popular choices include free-content licenses (CCs) and model-specific licenses.

The underlying reasons for the diverse mix of license types (OSS, free-content, model-specific) are beyond the intended scope of this work; some related discussion can be found in (McDuff et al., 2024). Here, we offer a few hints based on our observations:

- ML models can be seen as derivatives of both code and data and are possibly released alongside them.

- Almost all OSS licenses adhere to a non-discriminatory spirit, and oppose imposing behavioral restrictions, such as non-commercial use of materials.

- Neither OSS nor free-content licenses contain governing clauses toward generated content, which is a common product in model usage.

- Using OSS or free-content licenses for models may lead to misinterpretation of the license terms, causing confusion in selecting licenses for derivatives.

As mentioned in Section 1, using OSS and free-content licenses to publish models can lead to a **License Mismatch** issue, as their license terms are not well-defined in the context of ML. We expand on this topic in Section 3.1.

### 2.2. How ML Models Are Reused?

The reason we explore the behaviors of model reuse is that licenses may have terms governing their usage and the resulting derivatives. Legal noncompliance may occur if these terms are violated, and in some cases, the license granted to users may be terminated. Therefore, it is important to study

how ML models are reused and whether these behaviors may trigger license violations.

Model reuse is now appreciable, as stated by (Jiang et al., 2023), and our up-to-date results shown in Figure 1 further support this. The most popular model reuse methods are Finetune and Adapter, as reflected by the dominant colors of the radial objects in Figure 1, with their end nodes representing the resulting derivatives. In the zoomed-in view, derivatives created by Quantization (Rajput & Sharma, 2024) can also be observed. Small clusters surrounding these radial objects represent models that are reused less frequently. Additionally, the blue lines forming a network between nodes indicate derivatives created through Merge. The subgraph view displays the 3-hop neighbor of Llama-3.2-3B-Instruct, representing its derivatives, with a total of 1,371 models and 1,437 dependencies. This leads us to the question: *Do these derivatives comply with the licenses of original works?*

### 2.3. How Does Legal Noncompliance Occur?

We continue using Llama-3.2-3B-Instruct to illustrate potential legal noncompliance in model reuse. Its license[10] removes clause 1.v of Llama2 license (see Section 1) and grants ownership of derivatives to the licensee, making it permissive and less prone to conflicts. Although this license does not object to applying a different license to derivative models, non-compliant licensing choices can still pose legal risks. We query the license distribution of direct derivatives of Llama-3.2-3B-Instruct, as shown in Figure 3. Among these derivatives, 53% retain the original model's license, 28.7% do not declare a license, and 12.8% are republished under OSS licenses. For illustration purposes in this case only, we assume that all these reuse behaviors trigger the *derivatives*-related conditions in licenses[11]. We can identify two potential legal noncompliance risks based on the derivatives' licenses:

**Risk 1: Copyleft-style Terms.** Llama3.2 license requires that the use of the model complies with the *Acceptable Use Policy for the Llama Materials*. This poses a problem for derivatives without a declared license, as their downstream users may remain unaware of such restrictions and face the risk of violations (especially when Llama3.2 model is merged into these derivatives, refer to Figure 8). In such cases, Llama3.2 license may be terminated under Clause 6 due to this unintentional breach. After the license is terminated, the licensor can pursue legal action for copyright

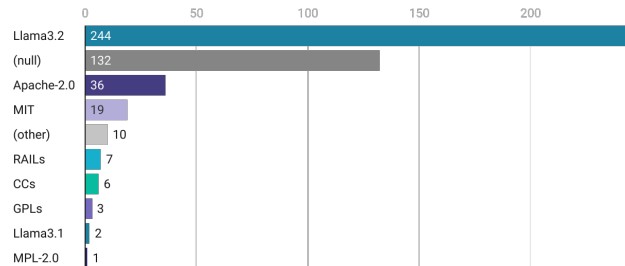

*Figure 3.* License Distribution of Direct Derivatives of Llama-3.2-3B-Instruct. Note that GPLs and MPL-2.0 are copyleft licenses.

infringement if the licensed work continues to be used[12].

**Risk 2: FSF-freedom violations.** It stem from combining works licensed under non-permissive terms with GPL-licensed works[13]. This noncompliance is also commonly observed in OSS projects (Kapitsaki et al., 2017; Ombredanne, 2020). Unfortunately, this risk has the potential to be one of the most common forms of noncompliance in ML projects due to the prevalence of behavioral use clauses in model-specific licenses (McDuff et al., 2024), and such clauses have been announced as incompatible with GPLs[14]. In fact, as of this paper's writing, no model-specific license has been approved by FSF or OSI[15]. This motivates the drafting of a new model license designed for general use and less prone to conflicts. We introduce this proposal in Section 3.4.

**Risk 3: Transitive Noncompliance.** It is worth noting that this marks only the beginning of noncompliance, not the end, as further legal risks emerge in indirect derivatives (derivatives of derivatives). For example, since GPLs are copyleft licenses, their derivatives must also be licensed under the same terms, which unfortunately amplifies noncompliance across all subsequent derivatives. Figure 7 in the appendix presents their license distribution, revealing that OSS licenses (38%) surpass Llama3.2 licenses (31%), reflecting increasing irregularity in license selection during model reuse and republication. Furthermore, some derivatives are licensed under Llama2 and Llama3, which contain exclusive terms that may conflict with Llama-3.2-3B-Instruct in a reverse manner. Finally, our current licensing practices can lead us to a stage, **a Quagmire of Legal Noncompliance**, where every project is in license violation, and everyone

---

[10]Llama 3.2 Community License Agreement

[11]For example, clause 5 of GPL-3.0, *Conveying Modified Source Versions*, will be triggered. In particular, the triggered conditions differ between licenses and may depend on the form of republication. See (Duan et al., 2024a) for further discussion.

[12]See CoKinetic Systems, Corp. v. Panasonic Avionics Corporation, 1:17-cv-01527, (S.D.N.Y.); Versata Software, Inc. v. Ameriprise Financial, Inc., 1:14-cv-00221, (W.D. Tex.).

[13]What does it mean to say a license is "compatible with the GPL?". See also Clause 7, *Additional Terms*, of GPL-3.0.

[14]"Various Licenses and Comments about Them" by Free Software Foundation (FSF), Inc.

[15]OSI Approved Licenses. We exclude *Blue Oak Model License* as it has no specific terms for ML models.

risks being sued. In the following sections, we further discuss and provide a quantitative preliminary analysis of the severity of legal noncompliance in ML projects.

## 3. Quantitative Analysis and Our Proposal

Our Llama-3.2-3B-Instruct case only presents a portion of the potential legal risks in model reuse. In this section, we provide a quantitative analysis of legal noncompliance, focusing on **License Mismatch**, **License Proliferation**, and **License Conflict**. At the end of this section, we will briefly introduce our draft model-specific license set, MGLs V2.0, as our proposal to mitigate these issues.

### 3.1. License Mismatch

As mentioned in Section 1, OSS and free-content licenses may not provide effective governance for model publishing, potentially leading to a situation where *You publish it, then you lose it* (Duan et al., 2024b). However, how much governance can these licenses actually offer, and where do they fall short? To address this, we design a table to quantitatively assess the clarity of licenses in the context of model publishing. As shown in Table 1, we assess license clarity across four main aspects: copyright license, patent license, ownership, and comprehensiveness. Each aspect is further divided into sub-items to measure whether the license clauses resolve (or declare) these elements.

We identified the following from the comparison:

- Model-specific licenses are better defined than OSS and free-content licenses, which often fail to resolve ownership disputes over derivatives and outputs.

- AGPL-3.0 and GPL-3.0 have the highest clarity among OSS licenses, however, as discussed in Section 2.3, they are prone to causing conflicts.

- Free-content licenses lack comprehensive provisions for patent declarations and grants.

- Both OSS and free-content licenses poorly govern remote model use, which is a common deployment mode for chat models.

- While most model-specific licenses are well drafted, they are not reusable, except for OpenRAIL-M and MGLs (including MG0-2.0 and MG-BY-∗-2.0).

However, we should emphasize that this clarity score is designed based on the model publishing scenario, and most OSS licenses and free-content licenses are well drafted for publishing their intended types of work. For example, CCs don't require patent license grants, as they are intended for publishing artwork. Overall, license mismatches can lead

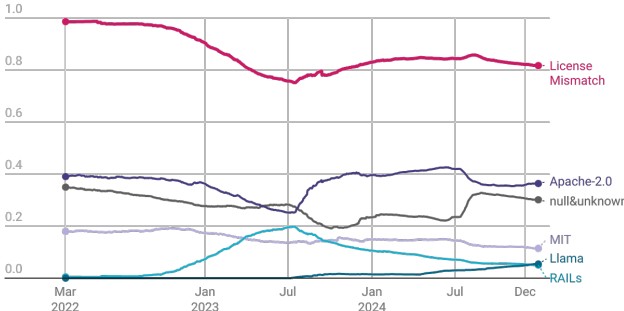

*Figure 4.* License Adoption Trends from Mar 2022 to Dec 2024.

to misinterpretation of terms and create legal ambiguity for users, potentially resulting in noncompliance.

The trends in license mismatches and adoption[16] are shown in Figure 4. The percentage of license mismatches is initially high but rapidly declines with the increased adoption of RAILs[17]. However, the mismatch rate rises again after July 2024 due to reduced RAILs adoption and increased Apache-2.0 usage. The proportion of models without declared licenses also grows, reflecting more ambiguity in licensing. Meanwhile, the recent slight drop in mismatches may be attributed to the explosive growth of Llama-related licenses. Overall, the mismatch rate remains high, consistently over 75%, highlighting the need for action from the ML community.

### 3.2. License Proliferation

Aside from the misinterpretation risk in license mismatch, publishing derivatives under a self-selecting license can also violate the copyleft terms of the original license (ref. Section 1). The scope of proliferation typically applies to "derivatives", which have different definitions depending on the license. In most cases, direct modifications of the original work that contain substantial parts of it are considered "derivatives" and should be subject to proliferation. On the other hand, whether methods like creating collections or linking (or via API) qualify depends on the specific case and varies across licenses. To enable analysis, we clarify the meanings of the four dependencies labeled by HF users[18]:

---

[16]Timestamps were collected using the HF Hub API via the *createAt* attribute. Repositories created before 2022-03-02 were assigned this date by HF. To illustrate the trends, repositories without a valid *createAt* value were excluded. *null* and *unknown* are considered mismatches, while *other* is excluded due to custom licenses are usually model-specific.

[17]HF's August 31, 2022 announcement: OpenRAIL: Towards open and responsible AI licensing frameworks.

[18]These self-reported dependencies may not conform to practice. See Appendix D for further discussion.

*Table 1.* List of Commonly Adopted Licenses & Agreements for Models on HF, Comparing License Clarity. Grouped by OSS, Free-Content (& Dataset), Model, and Sorted by Clarity Score. (including MGLs V2.0 Nov 2024)

| License Name | Copyright License | | | Patent License | | | Ownership | | Comprehensive | | | | | | | Clarity |
|---|---|---|---|---|---|---|---|---|---|---|---|---|---|---|---|---|
| | Decl. | Revo. | Subli. | Decl. | Revo. | Subli. | Derivate | Output | Rules | Remote | Def. | Warr. | Liab. | Term. | Gov. | |
| AGPL-3.0 | ✓ | ✓ | ✓ | ✓ | ✗ | ✓ | ✗ | ✗ | ≈ | ✓ | ✓ | ✓ | ✓ | ✓ | ✓ | 11.5 |
| GPL-3.0 | ✓ | ✓ | ✓ | ✓ | ✓ | ✓ | ✗ | ✗ | ≈ | ✗ | ✓ | ✓ | ✓ | ✓ | ✓ | 11.5 |
| AFL-3.0 | ✓ | ✗ | ✓ | ✓ | ✗ | ✓ | ✗ | ✗ | ≈ | ✓ | ✓ | ✓ | ✓ | ✓ | ✓ | 10.5 |
| OSL-3.0 | ✓ | ✗ | ✓ | ✓ | ✗ | ✓ | ✗ | ✗ | ≈ | ✓ | ✓ | ✓ | ✓ | ✓ | ✓ | 10.5 |
| LGPL-3.0 | ✓ | ✓ | ✓ | ✓ | ✗ | ✓ | ✗ | ✗ | ≈ | ✗ | ✓ | ✓ | ✓ | ✓ | ✓ | 10.5 |
| Apache-2.0 | ✓ | ✓ | ✓ | ✓ | ✓ | ✗ | ✗ | ✗ | ≈ | ✗ | ✓ | ✓ | ✓ | ✓ | ✗ | 9.5 |
| ECL-2.0 | ✓ | ✓ | ✓ | ✓ | ✓ | ✗ | ✗ | ✗ | ≈ | ✗ | ✓ | ✓ | ✓ | ✓ | ✗ | 9.5 |
| Unlicensed | ✓ | ✓ | ✓ | ✓ | ✓ | ✓ | ✗ | ✗ | ≈ | ✗ | ✗ | ✓ | ✓ | ✓ | ✗ | 9.5 |
| Artistic-2.0 | ✗ | n/a | n/a | ✓ | ✗ | ✓ | ✗ | ✗ | ≈ | ✓ | ✗ | ✓ | ✗ | ✓ | ✗ | 6.5 |
| MIT | ✓ | ✗ | ✓ | ✗ | n/a | n/a | ✗ | ✗ | ≈ | ✗ | ✗ | ✓ | ✓ | ✗ | ✗ | 4.5 |
| GPL-2.0 | ✗ | n/a | n/a | ✗ | n/a | n/a | ✗ | ✗ | ≈ | ✗ | ✗ | ✓ | ✓ | ✗ | ✗ | 3.5 |
| LGPL-2.1 | ✗ | n/a | n/a | ✗ | n/a | n/a | ✗ | ✗ | ≈ | ✗ | ✗ | ✓ | ✓ | ✗ | ✗ | 3.5 |
| BSD-3/2-Clause | ✗ | n/a | n/a | ✗ | n/a | n/a | ✗ | ✗ | ≈ | ✗ | ✗ | ✓ | ✓ | ✗ | ✗ | 2.5 |
| WTFPL-2.0 | ✗ | ✗ | ✗ | ✗ | ✗ | ✗ | ✗ | ✗ | ✗ | ✗ | ✗ | ✗ | ✗ | ✗ | ✗ | 0 |
| ODC-By-1.0 | ✓ | ✗ | ✓ | ✓ | n/a | n/a | ✗ | ✗ | ✓ | ✓ | ✓ | ✓ | ✓ | ✓ | ✓ | 10.0 |
| PDDL-1.0 | ✓ | ✓ | ✓ | ✓ | n/a | n/a | ✗ | ✗ | ✓ | ✗ | ✓ | ✓ | ✓ | ✓ | ✓ | 10.0 |
| CC-BY-∗-4.0 | ✓ | ✓ | ✓ | ✓ | n/a | n/a | ✗ | ✗ | ≈ | ✗ | ✓ | ✓ | ✓ | ✓ | ✓ | 9.5 |
| CC0-1.0 | ✓ | ✓ | ✓ | ✓ | n/a | n/a | ✗ | ✗ | ✓ | ✗ | ✗ | ✓ | ✓ | ✓ | ✗ | 8.0 |
| GFDL-1.3 | ✗ | n/a | n/a | ✗ | n/a | n/a | ✗ | ✗ | ≈ | ✗ | ✓ | ✓ | ✓ | ✓ | ✗ | 4.5 |
| C-UDA | ✗ | n/a | n/a | ✗ | n/a | n/a | ✗ | ✗ | ≈ | ✗ | ✓ | ✓ | ✓ | ✗ | ✗ | 3.5 |
| LGPLLR | ✗ | n/a | n/a | ✗ | n/a | n/a | ✗ | ✗ | ≈ | ✗ | ✗ | ✓ | ✓ | ✓ | ✗ | 3.5 |
| ◇ **MG0-2.0** | ✓ | ✓ | ✓ | ✓ | ✓ | ✓ | ✓ | ✓ | ✓ | ✓ | ✓ | ✓ | ✓ | ✓ | ✓ | **15.0** |
| ◇ **MG-BY-∗-2.0** | ✓ | ✓ | ✓ | ✓ | ✓ | ✓ | ✓ | ✓ | ✓ | ✓ | ✓ | ✓ | ✓ | ✓ | ✓ | **15.0** |
| † AI2-ImpACT-LR | ✓ | ✓ | ✓ | ✓ | ✓ | ✓ | ✓ | ✗ | ✓ | ✓ | ✓ | ✓ | ✓ | ✓ | ✓ | 14.0 |
| † AI2-ImpACT-MR | ✓ | ✓ | ✓ | ✓ | ✓ | ✓ | ✓ | ✗ | ✓ | ✓ | ✓ | ✓ | ✓ | ✓ | ✓ | 14.0 |
| † AI2-ImpACT-HR | ✓ | ✓ | ✓ | ✓ | ✓ | ✓ | ✗ | ✗ | ✓ | ✓ | ✓ | ✓ | ✓ | ✓ | ✓ | 13.0 |
| ◇ OpenRAIL-M | ✓ | ✓ | ✓ | ✓ | ✓ | ✗ | ✗ | ✓ | ✓ | ✓ | ✓ | ✓ | ✓ | ✓ | ✗ | 12.0 |
| † Llama3/.1/.2/.3 | ✓ | ✗ | ✓ | ✓ | ✗ | ✓ | ✓ | ✗ | ✓ | ✓ | ✓ | ✓ | ✓ | ✓ | ✓ | 12.0 |
| † Llama2 | ✓ | ✗ | ✓ | ✓ | ✗ | ✓ | ✓ | ✗ | ✓ | ✗ | ✓ | ✓ | ✓ | ✓ | ✓ | 11.0 |
| † OPT-175B | ✓ | ✓ | ✓ | ✗ | n/a | n/a | ✗ | ✗ | ≈ | ✗ | ✓ | ✓ | ✓ | ✓ | ✓ | 9.5 |
| † Gemma | ✗ | n/a | n/a | ✗ | n/a | n/a | ✓ | ✓ | ✓ | ✓ | ✓ | ✓ | ✓ | ✓ | ✓ | 9.0 |

**Header Definitions:**

**Decl**are: ✓ The license explicitly declares that a copyright/patent license (or a waiver thereof) is granted or reserved to the licensee. ✗ No explicit claim of granting a copyright/patent license.
**Revo**cability: ✓ The revocability of the granted copyright/patent license (or a waiver thereof) is explicitly stated; ✗ No explicit statement on revocability.
**Subli**cense: ✓ The license explicitly states if the granted copyrigh/patent license (or a waiver or auto-licensing term thereof) is sublicensable; ✗ No explicit statement on sublicensing.
**Derivative**: ✓ The ownership of IP rights in "derivative" works is explicitly resolved; ✗ The ownership of IP rights in "derivative" works is unclear, or "derivative" is not explicitly defined.
**Output**: The ownership of outputs is explicitly resolved; ✗ The ownership of outputs is unclear.
**Rules**: ✓ The license terms can cover common ML activities; ≈ Some ML activites fall outside the definition of this license; ✗ Almost no rules are set forth.
**Remote**: ✓ The license addresses remote access situations (e.g., via web services); ✗ No definitions or rules regarding remote access behaviors are set forth.
**Def**inition: ✓ The license includes definitions to clarifies key terms; ✗ No definitions are specified.
**Warr**anty: ✓ The license specified a disclaimer limiting warranties (or a waiver thereof); ✗ No warranty disclaimer is specified.
**Liab**ility: ✓ The license specifies the limitation of liability (or a waiver thereof); ✗ No limitation of liability is specified.
**Term**ination: ✓ The license specifies under what circumstances this license may be terminated or if it is never terminated; ✗ No termination conditions are specified.
**Gov**erning Law: ✓ The license specifies the governing law of this license; ✗ No governing law is specified.
**Clarity**: ✓ +1.0, ≈ +0.5, ✗ +0, n/a: +0 . Maximum Clarity Score: **15**.

† These licenses are not considered reusable as they contain proprietary copyright statements and include terms that structurally favor the licensor over the licensee.
◇These licenses are for public ML model publishing and are continuously updated; table items may vary with future versions and amendments.

**Finetune**: Represents a direct modification of the original model, e.g., full parameter finetuning.

**Adapter**: Represents an addon that relied on the original work but does not include any part of it, e.g., LoRA.

**Quantization**: Represents a transformed copy of the original work by compressing numerical representations.

**Merge**: Represents a deep fusion of models that is hard to separate, e.g., Weighted Averaging (Goddard et al., 2024).

Based on the above clarifications, we consider Finetune, Quantization, and Merge to constitute "derivatives" of the original work and, therefore, should be covered under license proliferation. We consider two situations: copyleft licenses and licenses with copyleft-style terms. For copyleft licenses, derivatives must be published under the same license; otherwise, a violation occurs, leading to license termination. In copyleft-style terms, the use of original models

and their derivatives must comply with specified conditions (e.g., use for non-commercial purposes). Adopting a different license for any works (including Adapter) based on the original increases the risk of unintentional violations in subsequent uses. The actual list of licenses in each category is provided in Appendix B, and the percentage of such noncompliance is shown in Figure 5.

Violations of copyleft licenses refer to the percentage of derivatives (both direct and indirect) that adopt a different license or *null*. As shown in Figure 5, the percentage of this violation is quite low (about 0.24%) across all repositories. A possible reason is that developers are more familiar with these copyleft licenses, which are commonly seen in OSS projects (Almeida et al., 2017) and legal precedents[12]. However, for licenses with copyleft-style terms, a growing trend in license changes was observed, ultimately settling at 17.7%. This highlights the need for action to raise awareness

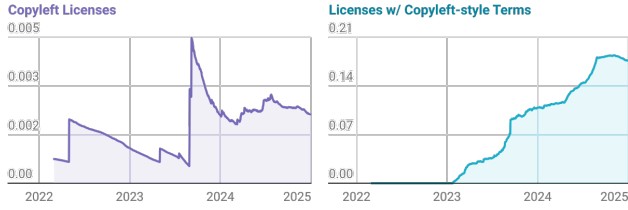

*Figure 5.* Trends in Violations of Copyleft License Proliferation and License Changes from (Licenses with) Copyleft-Style Terms.

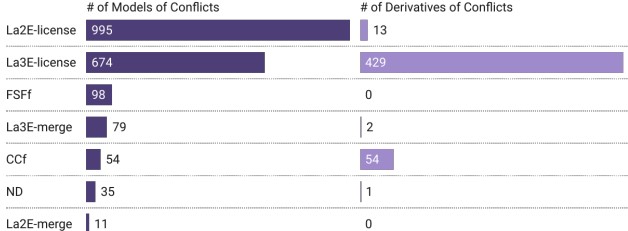

*Figure 6.* Models with License Conflicts and Their Derivatives.

of such risks. Note that the likelihood of these violations is much higher for qualified models (e.g., 63.5% for copyleft-style terms); refer to Table 3 in the appendix for further evaluation. Another risk of these copyleft-style terms is that they may lead to conflicts with the licenses of derivative works, a topic we expand on in the next section.

### 3.3. License Conflict

Even though the original license allows the licensee to adopt a new license for their derivatives, the chosen license may still be noncompliant with the original one due to conflicting terms (Chestek, 2024). For example, it would be noncompliant to finetune a OpenRAIL-licensed model and then republish it under GPL-3.0, as OpenRAIL includes behavioral usage restrictions that violate FSF's definition of freedom[13]. Note that this differs from the copyleft issue we mentioned above, which causes noncompliance in a reverse direction. CCs include terms to ensure downstream compliance with their defined freedom and are one-way compatible with GPL-3.0 in the case of CC-BY-SA-4.0[19].

Another conflict can arise from prohibited usage. For example, users cannot share derivatives if the original model is licensed under CC-BY-ND. Similarly, merging two models licensed under Llama2 and Llama3, respectively, is prohibited, as Clause 1.v of both licenses restricts using licensed models (and their outputs) to improve other models, except for themselves and their derivatives. The license conflicts targeted for detection in this work are listed in Table 2. In practice, there may be additional types of conflicts not covered in Table 2. For instance, reusing a model with a non-commercial license (e.g., CC-BY-NC) to create a proprietary work, or reusing a GPL-licensed model and refusing to disclose the source. However, it is difficult to determine whether an ML project is used for commercial purposes or if the source code has the appropriate release. Therefore, we focus only on the four conflicts that can be easily identified from the licenses and dependencies of models.

The count of distinct models with conflicts is presented in Figure 6. We observe that Llama2 and Llama3 licenses sig-

nificantly cause more conflicts compared to others. We apply two detection cases: 1) *License*: each model published under the Llama2/3 license must have Meta's Llama2/3 as its base model; otherwise, a conflict is reported. 2) *Merge*: a model can only merge Meta's Llama2/3 with Meta's Llama2/3 or their derivatives; otherwise, a conflict is reported. Note that reuse to create an adapter will also trigger analysis under the *license* case. This is because the Llama2/3 exclusive clause covers all behaviors involving the use of Llama materials, outputs, or results, not just derivatives. Since models that are fine-tuned, adapted, or quantized from Llama2/3 are naturally considered derivatives of Llama2/3 and thus comply with this exclusive clause, we only need to consider the conflict arising from *merge*.

Any derivatives of these models with license conflicts also involve noncompliance, and their count is shown in Figure 6, with subsequent conflicts observed in both Llama2/3-licensed models. Regardless of the license adopted, directly merging Llama2/3 models with non-Llama2/3 models constitutes noncompliance, as shown in La2/3E-merge. Fortunately, the resulting models are seldom reused further, so transitive conflicts are rare. Similar results are observed in FSFf and ND conflicts. Despite these conflicts not necessarily being caused by merges, the conflict-causing licenses (e.g., GPL-3.0, CC-BY-ND-4.0) are well-known as non-permissive and lead to fewer attempts to reuse. It is worth emphasizing that 95% of FSFf conflicts are caused by the copyleft-style terms in model-specific licenses, which are not typically observed in traditional OSS projects. Lastly, the CCf conflict is less common, but it can cause more transitive conflicts in derivatives.

Overall, 2.2% of models in our database with licenses and edges exhibit detected license conflicts (Figure 8 visualizes all noncompliance we identified in this work). Notably, this percentage represents distinct models, while the total count of conflicts is higher due to multiple occurrences within a single model. In some conflicts, the conflict ratio can reach 56.6% among qualified models (refer to Table 3 in the appendix). Our results reveal underlying legal compliance issues in using proprietary licenses like Llama2 and Llama3. In fact, only OpenRAILs are model-specific licenses for

---

[19]Understanding Free Cultural Works; Compatible Licenses

public use; however, the copyleft-style terms they share can, and have already, led to FSFf conflict, as we have shown. This motivates us to draft a new model-specific license that is less prone to causing conflicts, as we introduce following.

### 3.4. Our Proposal: ModelGo Licenses

We propose a set of model-specific licenses, named MGLs[20], to promote more standardized model publishing. As shown in Table 1, MGLs offer good clarity and, most importantly, are reusable licenses not bound to any proprietary models. Furthermore, MGLs have two features to promote better model licensing:

**License Elements**: Inspired by CCs, MGLs include five license elements to address various needs in model licensing: BY (Attribution), NC (Non-commercial), ND (No Derivatives), RAI (Responsible AI), and OS (Open Source). For example, MG-BY-NC-ND-2.0 indicates that the licensee must: 1) BY – Retain the original attribution; 2) NC – Prohibit commercial use of the licensed materials, derivatives, and outputs; 3) ND – Prohibit downstream users from distributing any derivatives or generated content. We also offer the highly permissive MG0-2.0 license without any elements. Rather than enforcing responsible behavioral restrictions through the license, we simply provide this option for developers to choose. We envision MGLs as a unified solution for model licensing and have submitted MG0-2.0, MG-BY-2.0, and MG-BY-OS-2.0 for license review by the Open Source Initiative (OSI). At the time of writing, several models have already adopted MGLs (Chen et al., 2025a;b).

**Model Sheet**: Inspired by MDL (Benjamin et al., 2019) and Model Cards (Mitchell et al., 2019), we advocate summarizing the grants and restrictions in license clauses into a sheet to reduce misinterpretation. For example, whether users are required to indicate modifications when distributing derivative materials. Additionally, our model sheet is useful for license selection and conflict analysis. An example model sheet for MG-BY-NC-2.0 is provided in Appendix Table 4. Our licenses are a complete reconstruction, collaboratively drafted with legal experts.

### 4. Alternative Views

**Behavior Use Clauses.** To prevent illegal and unethical use of models, some researchers advocate encoding such prohibitions into licenses (Contractor et al., 2022). These behavioral restrictions are widely adopted in current model-specific licenses (McDuff et al., 2024), such as RAILs, Llama2/3, Gemma. Some researchers advocate for ImpACT

---

[20]Version 2.0, Finalized on November 19, 2024, and continually updated based on feedback from developers and legal experts from the ML community and the OSI, please visit our website for the latest version. https://www.modelgo.li/

licenses (Zhao et al., 2024), which require users to submit derivative impact reports for licensor review. However, as shown in this work and by other researchers (Duan et al., 2024a), these additional restrictions make all subsequent derivatives incompatible with GPLs, potentially leading to legal noncompliance. Some argue that these terms may create barriers in the AI supply chain and misallocate responsibility to users (Klyman, 2024). Another viewpoint is that such restrictions can be ignored by malicious users and hinder model openness (Bommasani et al., 2024), prompting some developers to adopt Apache-2.0 for full openness (Liu et al., 2025b; La Malfa et al., 2024).

**Data Licenses.** Some studies focus on mitigating legal noncompliance by removing or unlearning non-permissive license data (Min et al., 2024; Kocetkov et al., 2023). Meanwhile, some efforts have been made to audit the provenance of data to identify lower-risk data for model training (Longpre et al., 2024). We acknowledge these efforts in promoting more standards and transparency in ML projects. However, the model license compliance issue remains an unresolved piece. To our knowledge, there are no data license terms that can be copylefted to models, and the choice of model license is independent of the license of training data. Furthermore, even if we promote resolving IP rights ownership through license clauses like MGLs, the copyrightability of model weights remains a controversial issue (Margoni, 2018).

### 5. Challenges and Opportunities

Figure 10 reveals a growing trend in model reuse, which exacerbates license compliance challenges. Nonetheless, the findings in this work represent only a small part of noncompliance issue and have certain limitations (see Appendix D). There are several promising research directions as opportunities. ModsNet (Wang et al., 2023; 2024a) constructs a knowledge graph that represents model dataset metadata and dependencies, enabling graph mining for noncompliance assessment. MLOps toolkits such as ModelDB (Vartak & Madden, 2018), MLflow (Zaharia et al., 2018), and Texera (Wang et al., 2024b; Huang et al., 2024) facilitate standardized management of ML workflows (Wang et al., 2025) and have the potential to improve transparency in model development and deployment. LicenseGPT (Tan et al., 2024) enables the evaluation of conflicts based on license text, serving as a valuable supplementary analysis tool for custom licenses and agreements. Some researchers have proposed encoding terms into formal rules for automatic noncompliance reasoning in an interoperable and extensible way (Zhao et al., 2021; Zhao & Zhao, 2024; Duan et al., 2024b), which could potentially be combined with the recently implemented AI Bill of Materials (BOM) standard (Bennet et al., 2024). However, AI BOM is not yet integrated into ML frameworks and HF platform, and

the self-reported model metadata is insufficient for comprehensive compliance analysis. ClearlyDefined (Vidal, 2023), DPI (Longpre et al., 2024), AI Alliance[21] are notable projects that are building transparency and compliance in the AI supply chain. From another perspective, research efforts such as GuardAgent (Xiang et al., 2024) and GuardReasoner (Liu et al., 2025a) aim to improve the intrinsic safety of LLMs, ensuring their alignment with human values and legal compliance.

## 6. Conclusion

The growing trends in model sharing and reuse, coupled with unstandardized model licensing, have catalyzed new legal compliance challenges in ML projects. These challenges are uniquely complex due to the presence of implicit AI supply chains and the interactions between multiple licenses. This study provides a novel quantitative analysis of license compliance issues in real-world model repositories, highlighting the urgent need for input from the ML community. The absence of standardized model licensing practices further motivates our proposal of a new set of model-specific licenses for general model publishing purposes. This study highlights the widespread license compliance issues in ML and outlines the path forward.

## Acknowledgements

This research/project is supported by the National Research Foundation Singapore and DSO National Laboratories under the AI Singapore Programme (AISG Award No: AISG2-RP-2020-018). Mengying Wang and Yinghui Wu are supported by NSF under OAC-2104007. Special thanks to Paul Lim Min Chim, Sheares Tiong, Eim Huiyeon, Rajah & Tann Asia and the NUS Office of Legal Affairs for their engagement and valuable comments during the drafting of ModelGo Licenses. Sincere thanks to Pamela Chestek for her professional and constructive insights during the review of the ModelGo Licenses.

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

## A. Disclaimer

The model licenses, along with any views, opinions, recommendations, or other information provided in this article, are intended solely for general informational purposes. The views and opinions expressed herein are those of the author(s) alone and do not represent any organization. This content does not constitute legal advice and should not be interpreted or relied upon as such. Readers are advised to seek professional legal counsel tailored to their specific circumstances and should not use the information provided herein as a substitute for such advice.

## B. Classification of Licenses

We classify the licenses used for model publishing on HF by following:

- **Copyleft License**: CC-BY-NC-SA-4.0, CC-BY-SA-4.0, GPL-3.0, AGPL-3.0, CC-BY-SA-3.0, GPL, OSL-3.0, GPL-2.0, Ms-PL, LGPL-3.0, MPL-2.0, EUPL-1.1, CC-BY-NC-SA-3.0, ODbL, CC-BY-NC-SA-2.0, CDLA-Sharing-1.0, LGPL-LR, LGPL, EPL-2.0, EPL-1.0, LGPL-2.1.

- **License with Copyleft-style Terms**: CreativeML-OpenRAIL-M, Llama2, CC-BY-NC-4.0, Gemma, Llama3, Open-RAIL++, Llama3.1, Llama3.2, OpenRAIL, BigCode-OpenRAIL-M, CC-BY-NC-3.0, BigScience-BLOOM-RAIL-1.0, Llama3.3, BigScience-OpenRAIL-M, CC-BY-NC-2.0, DeepFloyd-IF-License, C-UDA.

- **No Derivate License**: CC-BY-NC-ND-4.0, CC-BY-ND-4.0.

Copyleft licenses include terms that require derivatives of the licensed materials to be published under the same license. Copyleft-style terms refer to any restrictions that apply to subsequent use, such as the non-commercial use restrictions in CC-BY-NC-4.0 or the use behavior restrictions in RAIL and Llama licenses. No derivate licenses include terms that prohibit any form of sharing of derivatives. Note that all licenses listed in this appendix are those present in the database, and not an exhaustive set of qualified licenses.

## C. Supplementary Results

**Figure 7** illustrates the license distribution of indirect derivatives of Llama-3.2-3B-Instruct, where Apache-2.0 has surpassed Llama-3.2 as the dominant license, highlighting the intensified license mismatch. Additionally, some derivatives have adopted less permissive licenses, such as GPLs and Llama2/3, which are more prone to conflicts, as shown in Table 2.

**Table 2** lists the four types of license conflicts targeted for detection in this paper No Derivate (ND) conflict can arise from licenses such as CC-BY-ND-4.0 and CC-BY-NC-ND-4.0. According to the conflict-causing clause, any attempt to share derivatives of the licensed material constitutes a violation. Re-

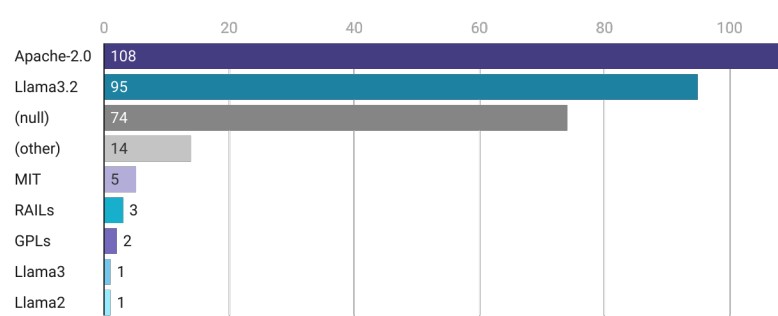

*Figure 7.* License Distribution of Derivatives of Derivatives from Llama-3.2-3B-Instruct.

call that adapters that do not contain substantial parts of the original work are not considered derivatives in this paper, consistent with common practices and real-world license terms.

The FSFf conflict arises when work under non-FSF-approved licenses (licenses that do not meet the FSF definition of freedom) is incorporated into a GPL-licensed project. This differs from a copyleft violation, which requires derivative works to be licensed under the same license (or a compatible license) as the original work. The list of non-FSF-approved licenses used in our experiment is: CreativeML-OpenRAIL-M, Llama-2, CC-BY-NC-4.0, Gemma, Llama-3, OpenRAIL++, Llama-3.1, Llama-3.2, CC-BY-NC-SA-4.0, OpenRAIL, BigCode-OpenRAIL-M, BigScience-BLOOM-RAIL-1.0, *other*, *unknown*, CC, CC-BY-NC-ND-4.0, Llama-3.3, CC-BY-NC-2.0, CC-BY-SA-3.0, Apple-ASCL, CC-BY-NC-3.0, CC-BY-2.0,

DeepFloyd-IF-License, PDDL, CC-BY-ND-4.0, CC-BY-NC-SA-3.0, Etalab-2.0, ODC-By, CC-BY-2.5, CC-BY-NC-SA-2.0, CDLA-Sharing-1.0, LGPL-LR, DeepFloyd-IF-License, ODbL, OSL-3.0, Ms-PL, EUPL-1.1, AFL-3.0. Since all model-specific licenses include behavioral restrictions, such as Acceptable Use Policies, they are all conflict-causing.

The CCf conflict can arise from adding new downstream restrictions to works under CCs (excluding CC0-1.0). Use restrictions in all model-specific licenses and copyleft restrictions in OSS licenses are considered violations of CC-freedom. The La2E and La3E conflicts arise from **using** LLaMA-related work to improve non-LLaMA work. This conflict is not license-related but depends solely on the model's dependencies, and creating adapters may also be conflict-causing.

*Table 2.* License Conflicts and Corresponding Conflict-Causing Clauses.

| Conflict Type (Abbrv.) | License | Conflict-Causing Clause |
|---|---|---|
| No Derivate (ND) | CC-BY(-NC)-ND-4.0 | Section 3 Clause a: *You do not have permission under this Public License to Share Adapted Material.* |
| FSF-freedom (FSFf) | GPL-3.0, AGPL-3.0 | Clause 10: *You may not impose any further restrictions on the exercise of the rights granted or affirmed under this License.* |
| CC-freedom (CCf) | CCs (excl. CC0-1.0) | Section 2 Clause 5B: *You may not offer or impose any additional or different terms or conditions on, or apply any Effective Technological Measures to, the Licensed Material if doing so restricts exercise of the Licensed Rights by any recipient of the Licensed Material.* |
| Llama2(/3) Exclusive (La2E/La3E) | Llama2(/3) License | Clause 1.v: *You will not use the Llama Materials or any output or results of the Llama Materials to improve any other large language model (excluding Llama 2(/3) or derivative works thereof).* |

**Table 3** presents the quantitative results of noncompliance detected in this work. The count of distinct models found to be noncompliant is marked as NCM, while the CTM for a noncompliance case represents the total number of qualified models that may either directly experience such noncompliance or be affected transitively through upstream dependencies. For example, all derivatives (both direct and indirect) of works under copyleft licenses contribute to CTM, whereas works under other licenses do not, as they cannot cause such noncompliance. Since all models can be affected by license mismatch, the CTM for this case equals the total number of models in our database. Similarly, because all derivatives result in ND conflicts, the CTM is equal to the NCM. The ratio of the count of NCM to CTM provides a better reflection of the likelihood of noncompliance occurrences, as shown in brackets.

To analyze which reuse methods are more prone to causing noncompliance, we classify models with noncompliance based on their adjacent reuse methods of the original models. The *n/a* indicates that noncompliance will not be caused by this type of reuse. For example, sharing adapters does not constitute the sharing of derivatives and is therefore not restricted by ND clauses. Note that the total count of the four reuse methods may be greater than NCM because some models may have multiple dependency paths from original models. The proportion of each reuse method is shown in the following brackets.

*Table 3.* Quantitative Results of Noncompliance. NCM: NonCompliance Models, CTM: Corrected Total Models.

| Noncompliance | # of NCM | # of CTM | Finetune | Merge | Quantization | Adapter |
|---|---|---|---|---|---|---|
| License Mismatch | 123,707 (81.8%) | 151,302 | *n/a* | *n/a* | *n/a* | *n/a* |
| Copyleft-style Terms | 26,679 (63.5%) | 41,019 | 11,011 (35.8%) | 1,453 (4.7%) | 4,775 (15.5%) | 13,546 (**44.0%**) |
| Copyleft Violation | 365 (38.5%) | 949 | 296 (**81.1%**) | 32(8.8%) | 37 (10.1%) | *n/a* |
| ND | 36 (100%) | 36 | 1 (2.8%) | 12 (33.3%) | 23 (**63.9%**) | *n/a* |
| CCf | 76 (2.1%) | 3,675 | 17 (20.0%) | 67 (**78.8%**) | 1 (1.2%) | *n/a* |
| FSFf | 98 (45.8%) | 214 | 84 (**85.7%**) | 0 | 14 (14.3%) | *n/a* |
| La2E-license | 1,000 (38.4%) | 2,602 | 182 (18.1%) | 29 (2.9%) | 331 (32.9%) | 465 (**46.2%**) |
| La2E-merge | 11 (55.0%) | 20 | *n/a* | 11 (**100%**) | *n/a* | *n/a* |
| La3E-license | 966 (24.7%) | 3,904 | 518 (**47.2%**) | 83 (7.6%) | 303 (27.6%) | 193 (17.6%) |
| La3E-merge | 81 (56.6%) | 143 | *n/a* | 81 (**100%**) | *n/a* | *n/a* |

Combining the results from this table and Section 3, we have the following findings:

- The likelihood of noncompliance based on CTM is notably high, particularly in the La2E-merge and La3E-merge cases, where more than half of the qualified models are deemed noncompliant.

- Some reuse methods are clearly more likely to lead to noncompliance, such as finetuning models under copyleft or

Llama3 licenses, finetuning models and then republishing them under GPLs, merging models under CCs, and making adapters of models under Llama2.

- The CCf conflict has the lowest likelihood, which may be attributed to CCs using license elements (e.g., BY, NC, SA) in their names to guide users on how to reuse the licensed material.

**Figure 8** visualizes the HF model dependencies and highlights detected noncompliance. Purple nodes represent **models without noncompliance**, while nodes of other colors indicate different types of noncompliance. Since models may suffer from multiple noncompliance issues, the display of nodes follows this priority: **La3E** > **La2E** > **FSFf** > **CCf** > **ND** > **Copyleft Violation** > **Copyleft-style Terms** > **License Mismatch**.

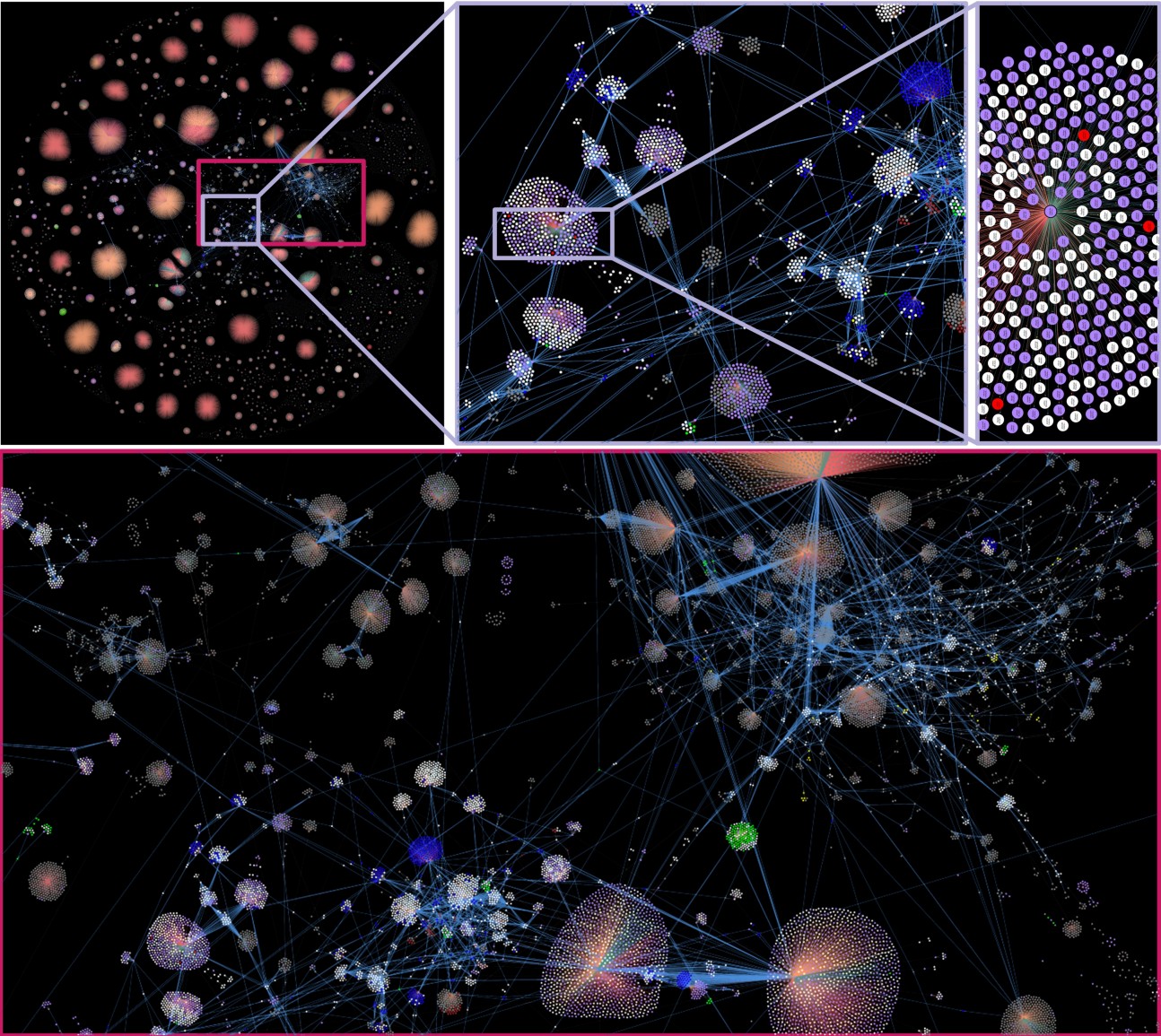

*Figure 8.* Visualization of Noncompliance in HF. Edge colors are consistent with Figure 1, and node colors, except purple, represent various types of noncompliance. The node view shows noncompliance in works related to meta-llama/Llama-3.2-3B-Instruct.

By comparing the global views of Figure 1 and Figure 8 (with differing layouts due to reruns), we can observe that many clusters are occupied by non-purple colors, indicating the occurrence of transitive noncompliance. The upper zoom-in

view focuses on the work cluster of Llama-3.2-3B-Instruct. By comparing it with Figure 1, we observed that it suffers significantly from Copyleft-style Terms issues, especially in works that involve merging. Additionally, a few FSFf conflicts were detected.

The lower zoom-in view presents the noncompliance in a merging-interlace area. Although merging accounts for only 4% of the total reuse methods, we found that it creates a large area of noncompliance, indicating the risks associated with these methods. This suggests that as model reuse becomes more frequent, noncompliance in ML projects will likely increase.

**Figure 9** further explores the causes of noncompliance related to meta-llama/Llama-3.2-3B-Instruct. The left image shows the node view with the **model ID**, while the right image shows the *license name*. We found that the three **FSFf** conflicts were caused by quantization of Llama-3.2, followed by republishing it under GPL-3.0 and AGPL-3.0 licenses. The *Acceptable Use Policy* does not align with GNU freedoms, rendering the new GPL licenses invalid. Since adapters are not considered derivatives, the Eltorio's model under AGPL-3.0 is not marked as an FSFf conflict. Violations of **copyleft-style terms** were detected in some derivatives due to the adoption of different or *null* licenses. The **La2E** and **La3E** conflicts are arising because Llama-3.2-3B-Instruct is not a derivative of the Llama2 or Llama3 models, respectively.

**Figure 10** illustrates the trends in model reuse frequency on HF. As observed, while recently growing reuse methods such as Finetune and Merge are slightly slower, but Quantization continue to grow steadily. Overall, the upward trajectory of model reuse frequency suggests an increased risk of legal noncompliance and license conflicts.

## D. Limitations

The main limitation of this work is that our model dependency analysis is based on crawled HF model tree metadata shared by users, which may contain misinformation. For instance, some repositories labeled as *Adapter* release the adapter file with the original model (either integrated or separate). This constitutes a modification of the original work rather than an independent model, and such cases should be included in license conflict analysis. On the other hand, there are repositories that publish adapters fine-tuned by LoRA but label them as *Finetune*. Additionally, some models labeled as *Quantization* actually involve full parameter fine-tuning after quantization, as indicated in their README files.

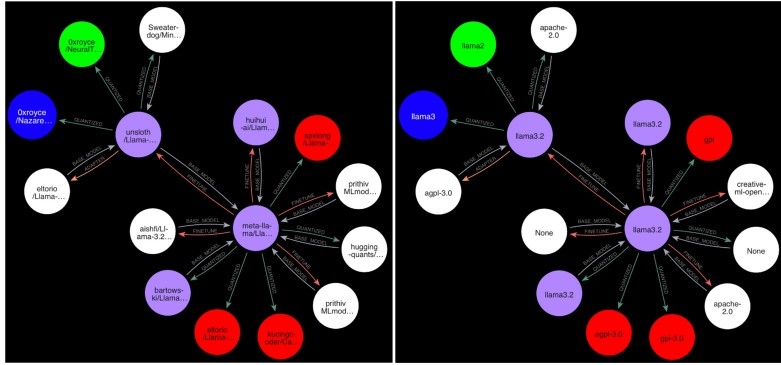

*Figure 9.* Node views of meta-llama/Llama-3.2-3B-Instruct and its derivatives with noncompliance issues. Left: Nodes labeled with model ID; Right: Nodes labeled with license names.

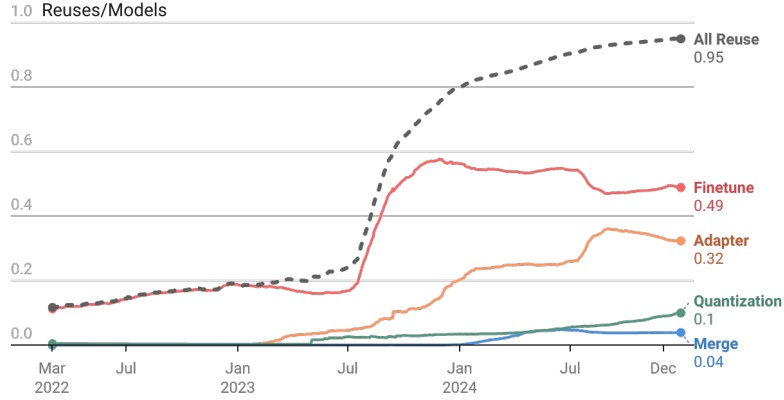

*Figure 10.* Trends in Model Reuse (Mar 2022-Dec 2024): The vertical axis represents the average reuse count (relationships) per model.

The metadata may also be incomplete, e.g., some license labels are not assigned by the publishers, but declarations can be found in their associated GitHub pages or academic publications. We also observe some repositories do not clearly mark their model trees, even though such dependencies are stated in their documents. Take our results in Table 3 as an example:

276 models have IDs containing the string *llama-3-∗* (case-insensitive), implying that their base model is likely Meta's Llama3, even though it is not explicitly stated in the model cards, leading to false positives in our La3E results. Similarly, 18 models indicate such dependencies in their model IDs in the La2E case. Additionally, HF records only four types of dependencies, excluding implicit dependencies such as those arising from distillation and tuning with LLM-generated data (Ren et al., 2024). Moreover, dependencies in proprietary models like Llama and Gemma are not disclosed by the publishers, leaving potential license conflicts within these widely reused models unknown.

Another limitation of our analysis lies in our database, which represents only a portion of the HF repository, specifically the top 8K models based on download counts along with their associated trees. We consider this strategy effective for filtering out rarely reused models and accelerating analysis and visualization. However, this breadth-first crawl approach may lead to some incompleteness in capturing model dependencies, potentially resulting in an underestimation of conflict evaluation outcomes.

## E. Model Sheet

Table 4 presents the Model Sheet for MG-BY-NC-2.0, a variant license of MGLs proposed in this work. These sheets outline the rights granted or reserved for the licensee and detail the requirements for preparing and publishing verbatim copies, derivatives, and generated content.

*Table 4.* Model Sheet of MG-BY-NC-2.0.

| | | |
|---|---|---|
| Grant of Rights to Licensed Materials | Use, Reproduce and Distribute Licensed Materials | ✓ |
| | Create Derivative Materials | ✓ |
| | Sub-License Licensed Materials | ✗ |
| | Revocable License to Licensed Materials | ✗ |
| | Ownership of Derivative Materials with Licensor | ✓ |
| | Commercial Use of Licensed Materials | ✗ |
| Grant of Rights to Derivative Materials | Use and Reproduce Derivative Materials | ✓ |
| | Distribute Derivative Materials | ✓ |
| | Sub-License Derivative Materials | ✗ |
| | Revocable License to Derivative Materials | ✗ |
| | Ownership of Derivative Materials with Licensee | ✗ |
| | Commercial Use of Derivative Materials | ✗ |
| Grant of Rights to Output | Right to Distribute Output if Indicate Notice of AI-Generated Content | ✓ |
| | Commercial Use of Output | ✗ |
| Responsible AI | Use Restrictions (RAI) on Licensed Materials, Derivative Materials and Output | ✗ |
| Requirements relating to Distribution of Licensed Materials and/or Derivative Materials | Provide a Copy of Original License | ✓ |
| | Retain Original Attribution Notice when Distributing Licensed Materials | ✓ |
| | Retain Original Attribution Notice when Distributing Derivative Materials | ✓ |
| | Indicate Modifications when Distributing Derivative Materials | ✓ |
| | Require Third Party Recipients to Comply with Use Restrictions (RAI) on Distributed Licensed Materials, Derivative Materials and Output | ✗ |
| | Disclosure of Distributed Licensed Materials and/or Derivative Materials in Source Code Form | ✗ |
| | Licensing Distributed Derivative Materials on Same Terms as License | ✓ |

