# OpenReview forum: "Position: Current Model Licensing Practices are Dragging Us into a Quagmire of Legal Noncompliance"
_ICML.cc/2025/Position_Paper_Track — ICML 2025 Position Paper Track oral_

### Official Review · Reviewer_UDyr · 2025-03-02

**Significance:** 3
**Argument Clarity:** 4
**Rating:** 3
**Confidence:** 3

**Questions:**

Please see Weaknesses.

**Discussion Potential:**

3

**Paper Summary:**

This paper discusses the unique challenges of licensing (derivative) ML models and examines common risks arising from improper license use. The authors conduct an in-depth analysis of current model licensing practices on Huggingface. Their findings validate the paper’s claim that existing licensing practices pose several risks of legal non-compliance.

The authors then discuss 15 essential attributes that a model publishing license should cover (Table 1). They show that most existing licensing options lack some of these attributes and subsequently introduce new licensing options that address all 15 attributes.

**Position:**

Yes

**Position In Title:**

Yes

**Related Work:**

3

**Strengths And Weaknesses:**

**Strengths**

1. The analysis of existing licensing practices effectively justify this paper's main claim: there is a "Quagmire of Legal Noncompliance".

2. The authors propose a new licensing option to help mitigate these issues.


**Weaknesses:

1. The paper lacks a dedicated section explaining which specific issues the newly proposed MGL licenses address. A section/table illustrating how MGL resolves the issues discussed in Section 3.1/3.2/3.3 would be helpful. For example, for the common use case A, using MGL instead of xxx license avoids issue Y.

2. The numerous abbreviations used in the paper can be confusing. While references are provided, it would improve readability if a cheat sheet were included in the appendix, briefly explaining key licensing options (e.g., those in Figure 3).


**Minor**

Typo: Line 18, "Trms" → "Terms"

**Support:**

4

---

> ### Author Rebuttal · Authors · 2025-03-31
>
> Thanks for the valuable comments.
>
> **How does MGL address legal noncompliance issues?** First, we want to clarify that MGL is not intended to resolve all the legal noncompliance issues mentioned in this work. Moreover, these issues cannot be fully resolved by a license alone, as we continue to see many cases of noncompliance with OSS licenses (e.g., `CoKinetic Systems, Corp. v. Panasonic Avionics Corporation, 1:17-cv-01527, (S.D.N.Y.) and Versata Software, Inc. v. Ameriprise Financial, Inc., 1:14-cv-00221, (W.D. Tex.)`). As mentioned in Section 3.4, MGLs aim to promote more standardized model publishing, which may help alleviate license mismatches if widely adopted. However, MGLs do not directly address license conflicts, which are primarily caused by the improper use of a license. For example, remixing models under MG-BY-OS and MG-BY-NC can lead to a license conflict. While we provide a compatibility table on our license website to prevent such issues, they may still occur. The main gap that MGLs aim to fill is the current lack of reusable licenses for public use when sharing models. Regarding legal noncompliance, we believe this is a complex challenge that requires cross-disciplinary efforts from policymakers, ML tool designers (e.g., AI supply chain management), and the ML community. As a position paper, the specific approaches to addressing the proposed challenges are somewhat beyond the scope, and we welcome debates and solutions from various perspectives.
>
> **Abbreviations used in this paper.** Thank you for your advice. The license abbreviations in Figure 3 follow a common practice, consistent with the license labels used in [Hugging Face](https://huggingface.co/models) or [SPDX license IDs](https://spdx.org/licenses/). We recognize that these abbreviations may be confusing to readers who are not familiar with the software/model licensing field, and we appreciate your suggestion to improve the readability of our paper. We will include an informative table in the appendix to address this.

---

### Official Review · Reviewer_PaSf · 2025-03-08

**Significance:** 4
**Argument Clarity:** 3
**Rating:** 4
**Confidence:** 4

**Questions:**

See above for discussion of overall goal.

Other minor questions:

- I wasn't totally clear on whether MGLs will use the same, or similar, restriction clauses from RAILS?
- Is the focus of MGL rollout to target individual developers on HuggingFace, or e.g. decision-makers in AI labs?
- Does the MGL approach take a stance as to how enforcement should be carried out?

**Discussion Potential:**

3

**Paper Summary:**

This paper takes the position that current model (i.e., "AI" model weight artifacts) licensing practices are creating present and future legal issues and argues that the ML community should adapt new practices that avoid high levels of noncompliance and ambiguity.

## update after rebuttal
I remain in favour.

**Position:**

Yes

**Position In Title:**

Yes

**Related Work:**

3

**Strengths And Weaknesses:**

Strengths:

- The draft has very strong motivation. Outside of the argumentation, I think the first few sections are just generally useful reading for anyone in the ML/AI space who hasn't engaged much with model licensing and model proliferation.
- Good framework of distinct issues here that can be clarifying for debates in literature and elsewhere (license mismatch, license proliferation, license conflict, implicit dependencies)
- There is very clear argumentation for the position before the supporting analysis
- The supporting analysis (empirical analysis of scraped HuggingFace data) relates directly to the position. I think this a reasonable amount of "empirical work" for a position paper, though others may disagree (and overall space in the paper leans more towards position than quantitative paper). The analysis is reasonable
- The proposed MGL approach is interesting, albeit a bit terse in terms of overall text dedicated to it (see below)

Weaknesses:

- The paper is a bit vague about how the proposed solution interplays with jurisdiction, enforcement in general, and especially pending enforcement attempts (that is, I think it's important to caveat a bit more heavily about how, unfortunately for many in the community who might like clear answers, there is a lack of clear reference cases for model license enforcement action).
- I think the work could benefit from a clearer statement of the overall normative goal of the work (i.e., the extent to which the goal is to promote model proliferation in general, to use licensing as a governance lever, to take a side in ongoing debates about open model weights more generally, etc.). I think the current draft is reasonable and don't think this is a "must have" -- but clearer positionality about the shared goals of the authors could be helpful in this kind of position paper, I think. Ultimately, this bullet point is very subjective.
- The overall space dedicated to the proposed MGL approach doesn't leave much room for discussion of concrete examples, which I think will be very helpful for this topic (there is an external link, which mostly solves this).

Ultimately, all these points above boil down to: this is a clear position, with good argumentation from prior work that also brings new data to bear.

**Support:**

4

---

> ### Author Rebuttal · Authors · 2025-03-31
>
> Thank you for the valuable legal insights.
>
> **For the difference between MGLs and RAILs.** We did not dedicate many pages to introducing MGLs, as it is somewhat outside the scope of a position paper. In fact, MGLs is the product of a one-year collaboration with a law firm and was drafted prior to this work. We have released full documentation for this license on our website and have promoted it through social channels, but due to ICML double-blind policy, we didn’t provide the link under this review procedure. MGLs is a completely new model license drafted by a law firm and does not copy terms from other licenses (in contrast to RAILs, which reuse terms from Apache-2.0). We provide a RAI option (e.g., MG-BY-RAI) for developers who favor responsible use terms. However, other license options from MGLs do not include such terms, while MG-*-RAI do not comply with Open Source Definition. Currently, MGLs is still being refined based on input from developers and lawyers. We have also submitted some variants (e.g., MG-BY) of MGLs for review by Open Source Initiative.
>
> **Is the focus of MGL rollout to target individual developers?** Sorry, we don't fully understand the differences in model publishing scenarios between individual developers and decision-makers in AI labs. However, one thing we do acknowledge is that MGLs is written with developers in mind, as it includes a ModelSheet to guide those who may lack the patience or legal knowledge to fully understand the license terms to choose the right one. However, the ModelSheet has been controversial, as some lawyers believe it could increase ambiguity in license interpretation. As a result, we are considering removing it from the license text in the next amendment. On the other hand, MGLs also offer licensing options like Non-Commercial (NC) which may appeal to decision-makers who wish to retain the rights to commercialize their models. We don’t claim that Open Source is better than Closed, MGLs simply provide a flexible solution and help people differentiate between these options.
>
> **Does the MGL approach take a stance on how enforcement should be carried out?** No, such provisions are not well-suited for a software license. In our opinion, a model license should primarily grant sufficient rights (i.e., patent rights, copyrights, database rights) for use and should avoid friction as much as possible. Maintaining legal compliance in the AI ecosystem will require cross-disciplinary efforts, including input from policymakers, AI workflow management, as well as the ML platform and community. Lastly, we believe MGLs' modularized license design can positively contribute to compliance analysis.

---

> > ### Comment · Reviewer_PaSf · 2025-04-02
> >
> > Thanks, these answers are very helpful!
> >
> > Just to clarify my question (no response needed and I think keeping this topic to the external documentation as you mentioned is reasonable): I meant to ask if MGLs are "intended" for e.g. grassroots "AI hobbyists" or individual researchers who might be publishing a small model on HuggingFace vs. something that aimed at organizations like Meta or OpenAI who are thinking about "open-ish" releases.

---

### Official Review · Reviewer_TKBe · 2025-03-13

**Significance:** 4
**Argument Clarity:** 3
**Rating:** 4
**Confidence:** 3

**Questions:**

- Are global legal differences take into consideration in the proposed solution? Different jurisdictions may have varying interpretations of licensing terms.
- Would the findings change if more models beyond HuggingFace models are included?
- What incentives or mechanisms would the authors propose to encourage adoption of MGLs by the industry and open-source ML communities?

**Discussion Potential:**

3

**Paper Summary:**

This paper discusses the challenges and legal risks associated with current machine learning model licensing practices. The authors argue that the lack of standardization in model licenses leads to legal noncompliance. They identify three main issues: license mismatch (applying software or content licenses that are unsuitable for ML models), license proliferation (inconsistent or conflicting license applications across derivatives), and license conflict (incompatibility between different licenses when models are merged or reused). The study includes an empirical analysis of over 15,000 models and 14,000 dependencies on Hugging Face, demonstrating widespread legal ambiguity. The authors propose a new set of model-specific licenses, MGLs, to address these issues.

**Position:**

Yes

**Position In Title:**

Yes

**Related Work:**

3

**Strengths And Weaknesses:**

**Strengths:**
- This paper studies how model licensing impacts legal compliance, which highlights a critical but often overlooked issue in the ML community.
- The study is comprehensive with detailed analysis using large-scale real-world data from HuggingFace.
- This paper is well-written and easy to follow, with a clear organization of licensing challenges into well-defined categories.
- The introduction of MGLs as a standardized licensing framework offers a constructive way forward, rather than merely identifying problems.

**Weaknesses:**
- While the paper highlights legal risks, it does not provide detailed legal case studies or expert-backed validation of how courts have handled similar disputes.
- The study primarily focuses on HuggingFace, which could introduce potential bias in data collection.
- The paper does not propose mechanisms for ensuring compliance with MGLs or preventing misuse of licenses.

**Support:**

3

---

> ### Author Rebuttal · Authors · 2025-03-31
>
> Thanks for your insightful comments.
>
> **For the jurisdictional concerns:** Our analysis in this work is based solely on license terms and does not take into account jurisdictional differences. As stated in the Appendix Disclaimer and Limitations, our analysis is merely a preliminary evaluation of potential legal risks in the current AI ecosystem and SHOULD NOT be construed as legal advice. In practice, whether a license violation has occurred requires verification by a court of law in different regions and depends on the context of specific cases. However, we do not believe it is a good criterion to ignore compliance with a model's license terms on the grounds that no one has been sued for violating these licenses yet. As shown in the OSS legal cases we reference in this paper (e.g., `CoKinetic Systems, Corp. v. Panasonic Avionics Corporation, 1:17-cv-01527, (S.D.N.Y.) and Versata Software, Inc. v. Ameriprise Financial, Inc., 1:14-cv-00221, (W.D. Tex.)`), violations of licenses can lead to lawsuits. Anything that can go wrong, will go wrong.
>
> PS: MGL includes the following clause to address jurisdictional issues. However, after review by third party, many lawyers believe this clause is problematic and could cause more issues than it resolves. As a result, we have decided to remove this clause in the next amendment.
>
> `7.	GOVERNING LAW AND DISPUTE RESOLUTION. Any legal action or proceeding arising out of or in connection with this License may be initiated only in the courts of the jurisdiction where the Licensor resides or conducts its primary business, and this License will be governed by the laws of that jurisdiction, excluding its conflict-of-law rules.`
>
> **For the findings change concerns:** It is possible. The most foreseeable change will be a dilution of noncompliance as we have omitted models that are less likely to be reused by others. As we mention in Appendix D Limitations, to speed up the analysis and focus on the major challenge, we only analyzed the top 8K most downloaded models and their related models. The remaining models, which are rarely reused and downloaded, are excluded, which are less likely to cause license conflicts. However, as shown in Figure 5 and Figure 10, there is a growing trend in both model reuse and license violations. Therefore, our position on the quagmire of legal noncompliance remains valid, as it is supported by our results.
>
> **For the adoption of MGLs concerns:** We are actively promoting MGLs. In fact, we have been preparing MGLs for over a year in collaboration with a law firm, and it was drafted prior to this article. We have launched the official MGL website and shared it across social channels (due to the double-bind policy, we are not providing the link here). Furthermore, we have submitted MGLs for review by the Open Source Initiative in pursuit of its approval, with the goal of making it the first open source model license. If MGLs gain OSI approval, they could serve as a strong incentive for ML developers who wish to claim their AI systems as open source. We are also encouraging adoption within the scope of our organization and envision AI conferences as a great platform to broadcast MGLs. Currently, we are still refining MGLs to ensure it gathers sufficient input from developers, lawmakers, and users.
>
> **For the misuse of MGLs concerns:** As a position paper, we have not dedicated many pages to introducing MGLs. We also acknowledge that it is impractical to prevent misuse of a license solely through its terms. In MGLs, we have designed the ModelSheet to inform users on how to choose and comply with these licenses. However, issues can still arise, such as remixing models under MG-BY-NC and MG-BY-OS. We want to emphasize that the primary goal of MGLs is to fill the gap created by the lack of a public, reusable model license for developers. Meanwhile, OSS and free-content licenses are not well-defined in the context of model publishing. We provide a compatibility table (in MGLs website) to resolve license conflicts between two MGLs, but more focus and effort will be needed to address the legal noncompliance issues highlighted in this work (e.g., the AI Bill of Materials). This is also a key motivation behind this paper.

---

### Official Review · Reviewer_2RM5 · 2025-03-13

**Significance:** 4
**Argument Clarity:** 2
**Rating:** 4
**Confidence:** 4

**Questions:**

How would you address the issue of derivation in ML model development, is it possible that, from the perspective of copyright laws, models are not strictly speaking derivatives of each other? How would this affect copyleft and copyleft-style licensing?

**Discussion Potential:**

3

**Paper Summary:**

The paper starts with an overview of issues in model licensing, focusing on license mismatch, proliferation, and conflict. It also introduces the concept of implicit dependency, due to inclusion of representations from other models. The paper then outlines approaches to ML model licensing and reuse, and based on this describes how legal noncompliance can occur. This is followed by a quantitative analysis, based on a large sample of models from HuggingFace. Based on this critical analysis, the authors propose a set of model-specific licenses that are meant to reduce the licensing problems outlined in the first part of the paper.

**Position:**

Yes

**Position In Title:**

Yes

**Related Work:**

3

**Strengths And Weaknesses:**

The paper address a key and interesting issue, related to model licensing. The overall analysis of challenges in model licensing is sound, and I appreciate that it is supported with quantiative data. At the same time, it mentions only in passing, and at the end, the fundamental problems related to copyrightability of model weights: an issue that lies at the heart of current challenges with model licensing, as there is no certainty that they can be meaningfully licensed. If model weights are not copyrightable, then the critical analysis presented in the paper, based on assumptions about how licensing should work, is no longer valid. Similarly, authors only briefly consider the complicated issue of derivation of models, when they introduce the idea of implicit dependency - again, it is a key assumption that should be investigated, as it has significant impact on issues related to license conflict, described in the paper. This is most problematic in the section on copyleft-style terms, where it is not clear how copyleft-style terms would apply in various model derivation processes.
Some parts and arguments in the paper are not very clear. The section on License Mismatch is very brief and provides one, not very convincing, example. The part on LIcense Proliferation seems to conflate the issue with that of derivatives (these are two different issues). Parts of the paper (for example, treatment of Table 1 or Figure 5) are very dense and brief. With limited space available, it would be helpful if the argument was simplified by removing some of the parts and focusing on the core of the argument.
Finally, it is not fully clear how the MGL solves the problems presented in the paper, especially that these are presented as having to do less with licensing terms, and more with application of various licenses.

**Support:**

3

---

> ### Author Rebuttal · Authors · 2025-03-31
>
> Thanks for your insightful comments.
>
> **For the model copyrightability concerns:** I agree that a key issue in current model licensing is the copyrightability of model weights, and whether ML training constitutes fair use [1] remains a controversial topic related to this. However, copyright is not the only IP rights embedded in an ML model, it may also involve patent rights or database rights (PS: Initially, MGL did not grant database rights, but some argue that such rights should be granted for the use of an AI model or system. In response, we are amending MGL to address this gap). Even if the copyrightability issue requires further legal justification from courts and lawmakers, license conflicts still exist. For example, the Llama 2 exclusivity conflict arises from Clause 1.v: `“You will not use the Llama Materials or any output or results of the Llama Materials to improve any other large language model (excluding Llama 2 or derivative works thereof)”` This restriction is not based on model copyright but rather on patent rights, as its trigger condition is the *use of the Llama 2 model*. We acknowledge that if models are not copyrightable, conflicts caused by CCs may not apply, as these licenses are governed by copyright law. However, this does not weaken the position of our paper. Instead, it highlights that legal compliance issues require not just input from the ML community but also engagement in broader legal and policymaking discussions.
>
> **For the implicit dependency concerns:** As we point out in Appendix D, it is indeed a major limitation of our work that we only analyzed four types of recorded dependencies available on Hugging Face. We envision that implicit dependencies, such as distillation, could lead to a wide range of legal noncompliance issues. However, we currently lack efficient tools to identify these dependencies. We believe that as AI Bill of Materials standards like SPDX 3.0 gain support, more ML construction information can be stored, making it possible to track such dependencies. Nevertheless, the main focus of our paper is to highlight the serious situation where current casual licensing practices are increasingly causing legal noncompliance in the AI ecosystem.
>
> **How the MGL solves the problems presented in the paper?** MGL is just a first step toward standardizing model licensing, rather than a panacea for all legal noncompliance issues. We believe our modularized licensing approach provides support for license conflict analysis tools. However, to fully address legal noncompliance, it will require collaborative efforts from AI workflow management, lawmakers, and the ML community. MGL could still lead to legal noncompliance if used improperly. For example, remixing models under MG-BY-NC and MG-BY-OS could create conflicts. Our license website provides a compatibility table to resolve such cases, but as we have stated, our license is not intended to solve all noncompliance issues. The gap we aim to fill is the lack of a reusable, model-specific license (except RAILs, but they are homogeneous and OSD-incompliant) for developers to choose for general models sharing.
>
> **Regarding the issue of derivation in ML model development.** Thank you for pointing out this concern. We recognize that the concept of "derivation" needs to be more clearly explained in this paper. In this work, the determination of model derivative creating is defined according to license. For example, Gemma License defines derivation as: `“(i) modifications to Gemma, (ii) works based on Gemma, or (iii) any other machine learning model created by transferring patterns of the weights, parameters, operations, or output of Gemma to that model in order to make it perform similarly to Gemma.”` Based on this definition, the concept of "derivations" extends beyond copyright to include patent law (triggered by the use of the model). Therefore, actions such as finetuning, adapting, merging, and quantizing would all constitute derivative works, potentially triggering copyleft-style terms in the model license. However, we do not consider adapting will trigger a copyleft license, not because it does not constitute a derivative work, but because there are currently no model licenses that are truly copyleft. Our assumption in this work is based on the fact that no copyleft model licenses currently exist, and only model licenses contain copyleft-style terms. To enable effective license analysis, we have made reasonable simplifications in this work. We acknowledge that, in real-world cases, the judgment of derivative works (in the context of license) is more complex and may differ slightly from our approach here. For example, under GPL-3.0, `a compilation of a covered work with other separate and independent works` does not constitute a derivative.
>
> *[1] Henderson, P., Li, X., Jurafsky, D., Hashimoto, T., Lemley, M.A. and Liang, P., 2023. Foundation models and fair use. Journal of Machine Learning Research, 24(400), pp.1-79.*

---

> > ### Comment · Reviewer_2RM5 · 2025-04-02
> >
> > On patenting, why do you think Llama is patented, can you provide a reference? Patent law is not triggered by use, as you suggest, and there is no such thing as a license based on patent law. I am surprised that you are suggesting this.
> > Also, contrary to what you write, if something is not copyrightable, then licensing conflicts are irrelevant, because a user does not require a license to use the work. I agree that it's worthwhile developing proper licenses, but the caveat needs to be made.

---

> > > ### Author Response · Authors · 2025-04-02
> > >
> > > **Why do you think Llama is patented?**
> > >
> > > Apologies for the ambiguity in my previous rebuttal. The first point I want to clarify is that the "model" I referred to in my rebuttal does not solely mean "model weights" but rather the broader concept of "licensed materials" as defined in the context of the license. For example, according to its license, Llama 2 is defined as
> > >
> > > ```
> > > “Llama 2” means the foundational large language models and software and algorithms, including machine-learning model code, trained model weights, inference-enabling code, training-enabling code, fine-tuning enabling code and other elements of the foregoing distributed by Meta at ...
> > > ```
> > >
> > > And the "Licensed Materials" as defined in the MGL is
> > >
> > > ```
> > > “Licensed Materials” means, collectively, the Model and/or Complementary Materials Distributed made available by the Licensor under this License. For avoidance of doubt, the Licensed Materials do not include any data used for the purpose of training, pretraining and/or evaluating the Model.
> > >
> > > “Complementary Materials” means the source code and scripts used to define, run, load, benchmark and/or evaluate the Model, and prepare data for the purpose of training, pretraining, fine-tuning and/or evaluation of the Model, and any tutorials, operating manuals, user guides and/or documentation that guide users in using, operating, implementing and/or customising the Model.
> > >
> > > “Model” means the machine-learning constructs and/or assemblies licensed by the Licensor under this License, including any checkpoints, learned weights, parameters (including optimizer states) and the model architecture (if applicable).
> > > ```
> > >
> > > So the question is not merely whether model weights can be patented or copyrighted, but whether the AI-embedded software/systems as a whole can be patented or copyrighted. We acknowledge that this is a controversial topic in both the legal and ML fields today. As we can observe, there is significant divergence in the rights granted across existing model licenses/agreements:
> > >
> > > - Llama2 license: You are granted a non-exclusive, worldwide, non-transferable and royalty-free limited license under Meta’s **intellectual property or other rights** owned by Meta embodied in the Llama Materials …
> > > - OpenRAIL-M: Both **copyright and patent** grants apply to the Model, Derivatives of the Model and Complementary Material.
> > > - OPT-175B License: … limited license under Meta’s **copyright** interests to reproduce, distribute, and create derivative works of the Software solely for your non-commercial research purposes.
> > >
> > > We agree that there is no evidence suggesting that model weights themselves can be patented, but the methods for using and creating models can indeed be patented (which may fall under "licensed materials"), as evidenced by:
> > > - Systems and methods for language model-based text insertion (US11886826B1)
> > > - Systems and methods for using contrastive pre-training to generate text and code embeddings (US20240370779A1)
> > >
> > > **No such thing as a license based on patent law.**
> > >
> > > As ML researchers with some collaboration with lawyers under our respective jurisdictions, we are not fully familiar with patent law worldwide. However, what we want to highlight is that many open source software licenses do grant patent licenses to licensees. Some examples include: `Apache-2.0, AFL-3.0, GPL-3.0, Artistic-2.0, OSL-3.0, ECL-2.0`.
> > >
> > > **If something is not copyrightable, then licensing conflicts are irrelevant, because a user does not require a license to use the work.**
> > >
> > > We agree that the copyrightability and patentability of models is an urgent issue that requires resolution from a legal perspective. However, as we mentioned previously, "model" does not just refer to "model weights". On the other hand, a software/model license may also involve contract law [1], as some licenses are binding contracts like:
> > > - Llama2 License: By clicking “I Accept” below or by using or distributing any portion or element of the Llama Materials, you agree to be bound by this Agreement.
> > > - MGL: By using, reproducing, Distributing, modifying and creating derivative works of the Licensed Materials (as defined below), You acknowledge and agree that You have read, understood, and agree to be bound by the terms and conditions of this License.
> > > - Gemma: By using, reproducing, modifying, distributing, performing or displaying any portion or element of Gemma, Model Derivatives including via any Hosted Service, (each as defined below) (collectively, the "Gemma Services") or otherwise accepting the terms of this Agreement, you agree to be bound by this Agreement.
> > >
> > > Therefore, even if we agree that model weights themselves may not be copyrightable and that copyright infringement may not be established, a material breach of the license terms could still constitute a breach of contract and is risky.
> > >
> > > [1] Chestek, P.S., 2024. A Promise without a Remedy: The Supposed Incompatibility of the GPLV2 and Apache V2 Licenses. Santa Clara High Tech. LJ, 40, p.303.

---

### Decision · Program_Chairs · 2025-04-30

**Decision:**

Accept (oral)

**Comment:**

This is a paper about how we tend to license ML models and why it is going to get us into trouble. The paper is well motivated, comprehensive, well written, and proposes a constructive solution to the problem. All four reviewers agree that the paper should be accepted.